# A Qualitative Study to Explain the Factors Influencing Mental Health after a Flooding

**DOI:** 10.3390/ijerph20010134

**Published:** 2022-12-22

**Authors:** Annika Hieronimi, Johanna Elbel, Michael Schneider, Inga Wermuth, Gerd Schulte-Körne, Dennis Nowak, Stephan Bose-O’Reilly

**Affiliations:** 1Institute and Clinic for Occupational, Social and Environmental Medicine, University Hospital, LMU Munich, 80336 Munich, Germany; 2Institute of Sociology, Ludwig-Maximilians-University Munich, 80801 Munich, Germany; 3Department of Child and Adolescent Psychiatry, Psychosomatics and Psychotherapy, University Hospital, LMU Munich, 80336 Munich, Germany

**Keywords:** extreme weather events, flooding, mental health, stressors, protective factors, children and adolescents

## Abstract

**Background:** Children and adolescents are considered to be particularly vulnerable to the psychological effects of climate change, such as extreme weather events. What are the protective factors and stressors for the mental health of the young population after extreme weather events in Germany? **Methods:** Nine semi-structured interviews with representatives of occupational groups providing care to children, adolescents, and political stakeholders were conducted in Simbach am Inn, a German town affected by flooding in 2016. The interviews were analyzed using qualitative content analysis according to Mayring. **Results:** The interviews show that the parents’ dealing with what they had experienced and the concern for their relatives were the most influential stressors for children and adolescents. As protective factors, they felt that conversations with familiar people and restoring a certain “normality” were particularly important. The interviewees described both, the time of the flooding, and the time after the initial state of shock had subsided, as particularly stressful. Consequently, the experts reported on children and adolescents acutely complaining of fear, helplessness, and extreme tension. Nevertheless, the demand for psychological care increased only slightly after the flooding in Simbach am Inn. **Conclusion:** The social environment of children and adolescents is essential for their psychological well-being after an extreme weather event. Research, especially on children and adolescents who have already been affected, must increase in order to be able to describe influencing factors even more precisely, to protect individuals from adverse mental health effects, and to identify healthcare requirements.

## 1. Introduction

Extreme weather events (EWEs) are becoming more intense and frequent worldwide as climate change increases [1]. By 2023, an estimated 175 million children and adolescents will be affected by the impacts of various disasters caused by EWEs [2]. In this regard, children and adolescents are among a particularly vulnerable population due to their limited life experience and few learned coping strategies [3].

In addition to personal factors, such as age, gender, and social background, event-related factors, such as being directly or indirectly affected, also play an important role in how mental health is affected by an EWEs [4]. Social factors, such as the mental health of primary caregivers or separation from parents after the event, are relevant as well [5,6].

Approximately 25% of children and adolescents show symptoms of posttraumatic stress disorder (PTSD), anxiety disorder, or depression after experiencing a EWEs, as described by the American Academy of Pediatrics [7]. Especially in younger children, psychological complaints may become somatic and appear, for example, as abdominal pain or manifest in behavioral changes [8,9].

Even before an EWE occurs, there are opportunities to protect the mental health of young people. Preventive interventions and educational activities can strengthen resources and thereby instill a sense of safety and self-efficacy [10].

In the aftermath of an EWE, the provision of “mental health first aid” is essential for young people to process what they have experienced [11]. However, at the current time, there is no suitable guideline for the psychosocial emergency care of children and adolescents after EWEs in Germany [12]. Moreover, involvement in action and provision of age-appropriate information are beneficial for the psychological well-being of the young population [13]. In addition, research shows that the restoration of everyday normalcy is also helpful for children and adolescents [14].

Even in Germany, EWEs are no longer a seemingly remote danger. On 1 June 2016, heavy precipitation from several thunderstorm parcels in the town of Simbach am Inn caused the level of the Simbach, a small stream of the same name, to rise rapidly. At midday, a road embankment collapsed under the weight of the water, causing a tidal wave to roll toward the city center. Five people died and several hundred people were rescued from houses and apartments. Approximately 110 pupils and their teachers had to persevere in the school center until the evening hours [15]. The inhabitants of the town had neither experienced a flood to this extent nor a flood due to the Simbach before.

With this background in mind, the following research questions are derived.

What mental effects were observed in connection with the flooding among children and adolescents in Simbach am Inn?What were the stressors respectively the protective factors for potential mental effects of EWE among children and adolescents in Simbach am Inn?How were the mentally affected children and adolescents cared for after the EWE in Simbach am Inn?

## 2. Materials and Methods

With the help of a semi-structured interview guide, representatives of the occupational groups providing care for children and adolescents, as well as political stakeholders, in Simbach am Inn were interviewed about how young people dealt with the flooding disaster.

### 2.1. Preparation of the Interview Guide

In cooperation with an interdisciplinary team of scientists from sociology, pedagogy, psychiatry, and medicine, a semi-structured interview guideline was created after extensive literature research. Two pre-tests were conducted on doctors to review the questions and interview duration. The final interview guideline included the questions in Table 1.

The ethics committee at the Medical Faculty of LMU Munich approved the study as ethically sound on 25 August 2021, with project number 21-0505. The study was also approved by the official data protection officer of the University Hospital of Munich (project 1691.a). The participants signed a data protection form before starting the interview.

### 2.2. Selection of Participants

The experts provided an indirect source of information about the mental health of children and adolescents in their care. They had an overview of the response and care of many young people. Interviewing caregivers and politicians instead of children and youth directly affected eliminated the risk of retraumatization.

Individuals who were either medical and therapeutic staff, school and educational staff, first responders, or representatives of the church and politicians and who were working in the care and support of children and young people at the time of the flooding were contacted. Inclusion and exclusion criteria (Table 2) were checked in a first phone call.

Of the fifty-two individuals contacted, nine agreed to be interviewed (Table 3). Interview cancellations occurred due to vacation or lack of interest. In addition, 17 contact attempts were left unanswered. Simbach is a small town of approximately 9500 inhabitants in 2016 from whom authors identified 52 potential participants who met inclusion and exclusion criteria. Planning the study, it was expected that, due to the limited number of interviewees, the study would function as an exploration of the subject and as a basis for further research.

### 2.3. Data Collection

A total of nine interviews were conducted between September and October 2021. Audio recordings of the interviews were made, transcribed, and pseudonymized. The interviews lasted between 20 and 65 min with an average of approximately 38 min. The transcript of the interview was sent to the respective interview partner for approval. Only after the interview and when partners agreed that the transcript was in order the coding was started. This methodology was discussed and optimized with several experts.

### 2.4. Data Analysis

The transcripts were analyzed with the help of MAXQDA 2020 (VERBI Software. Berlin, Germany) using qualitative content analysis according to Mayring. Deductive category formation was used to create individual categories in a theory-based manner before the interviews were conducted [16]. With the help of the stress-buffering model, the following categories were established [17].

Stress: stressors caused by the flooding event.Distress: mental impairments.Support: protective factors before, during, and after the flooding event.

Different codes, which were expected after reviewing the literature, were assigned to these categories. During inductive category building, the codes were expanded or reduced after reviewing the transcripts [16]. Based on the resulting final code system, the text passages of the individual interviews were assigned to different categories and then analyzed. Different experts compared their interpretation of codes and categories with the interviewer’s application of the codes and categories and discussed their respective views. By coding the interviews, clusters of themes were formed, and repetitions and patterns, as well as outliers, were identified. The results of the interviews serve to provide context and background and are, therefore, not subjected to theoretical triangulation.

## 3. Results

### 3.1. Stressors

One child wrote: “What’s happening to me, can I save myself? And mommy, what about my mommy?” (SCHUPA1)

According to the interviewees’ perception, the children’s concern for their parents was the predominant psychological stressor at the time of the flooding and, additionally, the most important emotion of the children and adolescents. Since parents and children were mostly in separate places at the time of the flooding, both children and parents were afraid for their relatives. This feeling was promoted by (social) media coverage of the event and a non-functional telephone network. In particular children and adolescents who had to reach safety on their own or were rescued were described as psychologically affected by the experts. Interviewees related this to individuals feeling overwhelmed during the event. The interviewees also described the loss of close people as a decisive factor for the psychological impact.

“Families were split up, the children with the aunt, the parents with their own parents, because you couldn’t find anything where everybody could be at the same time.” (MEDPSY1)

Interviewees reported that on the day of the flooding, many people in Simbach am Inn were forced to abandon their homes due to flood damage. With great solidarity, families were taken in by relatives, friends, or strangers. Since families were often separated due to capacity or had to live with many people for a period of time, there was no place of retreat where affected families could work through what had happened together.

“Children always care a lot about their parents. So, there was no hiding that.” (SCHUPA3)

As a result of the flood damage to homes and commercial spaces, some families experienced financial difficulties. Due to an uncertain financial future, families left the city and relocated. The interviewees described that children and adolescents adopted these existential worries of parents.

“If one family is done shovelling [mud] after three days, then it can be that on the fourth day they fall into this hole and say, how do we go on now?” (SCHUPA2)

Interviewees mentioned two points of time of greatest psychological stress: the time of the flooding, and the time when the physical exhaustion decreased and the processing of the experiences began. The latter varied widely across individuals in terms of timing and was related to the material impact of the flooding on the family.

### 3.2. Psychological Impairments

The acute reactions of children and adolescents to the flooding event were described as extreme fear, helplessness, and tension. One interviewee was able to explain specific age-dependent reactions and attributed the strongest impact to children between six and twelve years of age. In contrast, two interviewees described that children sensed the events exciting and subsequently re-enacted them playfully.

With a time-lag from the event, several respondents observed sleep disturbances and concentration difficulties in some children and adolescents. According to the interviews, even today, six years after the flooding, some young people in Simbach am Inn still struggle with recurring anxiety during heavy rain.

Despite all the impacts described, one interviewee also highlighted the resilience of the children and young people:

“I was surprised in terms of the stability of the children when they came [at the start of school about 10 days after the event].” (MEDPSY2)

### 3.3. Protective Factors

The interviewees rated the conversations about the experience in the private environment of the children and adolescents as the most important protective factor. In addition to parents, teachers and educators were also named as important contact persons. They paid attention to mental problems and acted as important intermediaries for help when needed. Children and adolescents took advantage of the low-threshold counseling offered by people they knew. In general, a stable and loving family environment was perceived as particularly important for the mental health of children and adolescents by the interviewees.

The interviewees also mentioned offers from aid organizations and political stakeholders, which, however, were directed more at parents than at children and adolescents themselves. In addition, some interviewees felt the need to adapt crisis intervention materials to natural disasters and children’s needs.

“The sooner the school, [the] association are back on track, that’s the most important. [...] After all, the children want the familiar, what they always do.” (MEDPSY2)

In addition, a familiar daily routine was described as very beneficial for the mental health of children and adolescents. Reopening schools and sports clubs were essential for their well-being.

“Many older children and young people [...] helped out with the clean-up and I think they were able to process a lot about that. They realized they could do something.” (MEDPSY1)

Adolescents were especially able to process what had happened by helping with the clean-up. This coping strategy was compared to that of grieving by two interviewees. In addition, the great solidarity and sympathy of the population were very helpful for both young people and adults. The resulting strong social network helps individuals to deal with emotions and provides them with material donations. Two interviewees observed that the focus was on the physical integrity of the relatives and that material items were perceived as secondary.

According to the assessment of one interviewee, the demand for psychological care increased slightly following the flooding. School psychologists were important contact persons for the children and adolescents affected. However, one respondent mentioned that due to the long waiting times, less complex mental illnesses could already be alleviated by talking to the parents or pediatrician. Due to the geographical location, medical and psychological care were always guaranteed in Simbach am Inn. However, it was hardly used in the first weeks after the flooding.

When it came to commemoration offers, the interviewees’ opinions varied widely. Some believed that the experience had been sufficiently dealt with in private and that no further public offerings were needed. Others declared themselves in favor of a memorial in the city as a reassurance that the event had occurred.

## 4. Discussion

The feelings and thoughts, especially of the children and adolescents still living at home, were strongly dependent on the parents’ concern and handling of the experienced flooding. This is in line with the research results from non-European countries mentioned in the background section [5,6].

In the interviews, the respondents mainly described short-lasting stress reactions, such as extreme fear, helplessness, and tension.

The need for adapted crisis intervention methods for children and adolescents after an EWE was expressed by experts in Simbach am Inn, as well as in the literature [12]. Offers to prepare for a possible EWE, for example, in the context of school lessons, were not mentioned by any of the interviewees in the city.

### 4.1. Interpretation

According to the interviewees, the main caregiver of children and adolescents is not only considered the greatest stress factor, but also the greatest protective factor for their mental health after the experience of an EWE.

The experts named concern for their parents and physical separation from them as the most important stressors. Hence, at least these two factors would be eliminated if children and adolescents could stay in one place with their parents during an EWE. Consistent with the viewpoint in the literature, being present and talking with parents were important for young people’s mental health [6,11].

This raises the question of whether emergency mental health services should be adapted to meet the needs of children and adolescents when they have been sufficiently helped by the low-threshold services provided by those they are familiar with. In this case, it would be more resonable to provide parents with information on how to deal with their children after an EWE and to train first responders in how to support parents. Karutz and Plagge already formulated the former in their 2018 recommendations for action for parents and other adult caregivers [12]. However, the prerequisite for psychosocial support for children and adolescents by their primary caregivers is that they have not been harmed physically, emotionally, or socially by the EWE.

### 4.2. Limitations

Several limitations can be identified in the research design. Only nine participants were interviewed, and saturation could not be reached. Therefore, the views present a baseline for further research. Due to the limited number of interviews, some perspectives cannot be presented and representativity is limited.

In Simbach, the medical infrastructure was not affected, so the possibility for professional help existed. This is not the case in other flooded regions. Further, other EWEs have different damage patterns in the area and cause other illnesses and injuries. The results are hence not transferable to all localities.

The interview participants were only able to respond to a limited extent to what the children and adolescents had experienced. They could only reproduce the information the young people shared with them about their feelings and worries and relied on the information to be as detailed as possible. Their own personal distress from the flooding was also reflected in the interviewees’ responses. Furthermore, the interviewees did not refer to specific psychiatric diagnoses; therefore, all statements given were subjective.

In order to be able to present the emergence and care of mental impairment after an EWE among children and adolescents in more detail, this age group should be directly interviewed. Due to ethics regulations in Germany and the risk of retraumatization, this was not possible.

## 5. Conclusions

The study shows that the direct social environment of children and adolescents was crucial for their handling and processing of the flooding in Simbach am Inn 2016. There is a need to expand research on the mental impact of EWEs on children and adolescents in affected communities in Germany, as climate change and related EWEs will increasingly burden society and healthcare systems in the future. In particular, the views of children, adolescents, and their parents must be explored to better understand the protective factors and stressors resulting from EWEs.

## Figures and Tables

**Table 1 ijerph-20-00134-t001:** Interview guideline.

Question Block	Questions
General	• What do you remember when you recall the event?
	• How affected by the flooding event were you or your social environment?
	• How did this affectedness manifest itself?
Psychological stress of children and adolescents	• How did children and adolescents in your professional or private environment perceive the flooding?
	• Were you able to detect differences in perception at different ages?
	• If you look back at the time during and after the flooding, at what point do you estimate the psychological stress of the children and adolescents was highest?
Care	• What was the medical and psychological care like at that time?
	• Who could children and adolescents turn to with their concerns and problems?
	• How can the population still think back to the event today?
	• Are there special services for children and adolescents?
Reflection	• Imagine that you have three wishes: What would you wish for in relation to the flooding of 2016?
	• Is there anything else you would like to share?

**Table 2 ijerph-20-00134-t002:** Inclusion and exclusion criteria.

Criteria		
Inclusion	General	• on-site in Simbach or the close surrounding area during the flooding
	Medical and therapeutic staff	• practicing in Simbach before and after the flooding
		• mainly working in the care of children and adolescents
	School and educational staff	• working at the school center in Obersimbach on the day of the flooding
	First aiders	• on duty during the flooding
	Representatives of the church and politicians	• in office on the day of the flooding
Exclusion	General	• no mental capacity to talk about the flooding
		• no presence in Simbach during the time of the interviews

**Table 3 ijerph-20-00134-t003:** Distribution of occupational groups of the interview partners.

Occupational Group	Interviews Conducted
Medical and therapeutic staff	2
School and educational staff	3
First aid providers	2
Representatives of the church	1
Representatives of politics	1

## Data Availability

Not applicable. The data are not available due to data protection regulations in Germany.

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
