# Peer review of "A Qualitative Study to Explain the Factors Influencing Mental Health after a Flooding"

_ijerph, 2022, doi:10.3390/ijerph20010134_

Round 1

Reviewer 1 Report

Thank you for the opportunity to review this paper. I found this paper interesting, and I think that the authors have performed interesting research that can be useful for researchers, parents, and psychosocial support workers. The primary concern is the small sample size, which probably affects the research results. Other than the sample size, the followings are my comments:

-line 54 ”Approximately 25% of children and adolescents …” Please add more details, for example, in one country or worldwide, what year.

-line 71 - line 80. I would suggest writing this paragraph concisely. 

-I would suggest adding previous research about EWE and the mental health of children and adolescents in the introduction section. Did you find any research gaps that helped you derive the research questions?

-Please add inclusion and exclusion criteria in the selection of participants.

-line 145 “... viewing the transcripts (16).” Please check “(16)”.

-line 259 “... in the background section (5, 6).” Please check “(5, 6)”.

-line 278 “.. for young people's mental health (6, 11).” Please check “(6, 11)”.

-line 286 “... and other adult caregivers (12).” please check “(12)”.

-I would suggest including the interview questions in the manuscript. Did the interview questions cover the research questions? 

-Did every participant answer the same questions? Were they open-ended questions? How long did each interview last?

-line 312 “One must learn to deal with the unavoidable and avoid the unmanageable.” (KIR1)” I would recommend authors double check if the sentence is appropriate in the conclusions section. 

-In the discussion section, I would suggest authors add the comparison of more previous research with this study. What are the new findings? Did this study find the answers to the three research questions?

Reviewer 2 Report

To explore the impact of extreme weather events on mental health of children and adolescents, this study conducted 9 semi structured interviews with representatives at a German town after encountering flooding. I recommend this MS to be published after a major revision.

Some suggestions for authors:

1.      The title is too long to reflect the the key point of this research.

2.      Does the EMEs or the disasters caused by EMEs affect mental health?

3.      I’m not sure if the descriptive approach of abstract of this manuscript is appropriate. I tend to use one paragraph that the context is logically connected to each other, to avoid repetitive description like the results and conclusion in this abstract.

4.      Is the number of interview partners enough? How to meet the randomness of interviewees? How to ensure the quality control?

5.      The manuscript should highlight that whether the results and conclusions deserved from this investigation are representative and applicative for other regions and other EMEs.

Round 2

Reviewer 1 Report

The authors have satisfactorily addressed all my concerns in my previous reviews. The revised manuscript is ready for publication.

Reviewer 2 Report

I think the authors have addressed my questions to some extent.